# The student becomes the master: Outperforming GPT3 on Scientific Factual Error Correction

**Dhananjay Ashok    Atharva Kulkarni    Hai Pham    Barnabás Póczos**
{dhananja, atharvak, htpham, bapoczos}@cs.cmu.edu
School of Computer Science, Carnegie Mellon University

## Abstract

Due to the prohibitively high cost of creating error correction datasets, most *Factual Claim Correction* methods rely on a powerful verification model to guide the correction process. This leads to a significant drop in performance in domains like scientific claims, where good verification models do not always exist. In this work, we introduce $\mathbb{S}$ci$\mathbb{F}$ix, a scientific claim correction system that does not require a verifier but can outperform existing methods by a considerable margin — achieving correction accuracy of 84% on the SciFact dataset, 77% on SciFact-Open and 72% on the CovidFact dataset, compared to next best accuracies of 7%, 5%, and 15% on the same datasets respectively. Our method leverages the power of prompting with LLMs during training to create a richly annotated dataset that can be used for fully supervised training and regularization. We additionally use a claim-aware decoding procedure to improve the quality of corrected claims. Our method outperforms the very LLM that was used to generate the annotated dataset – with Few-Shot Prompting on GPT3.5 achieving 58%, 61%, and 64% on the respective datasets, a consistently lower correction accuracy, despite using nearly 800 times as many parameters as our model.

## 1  Introduction

The widespread adoption of the Internet has led to the distribution of more written content than ever before in human history, and recent advances in generative AI models are expected to push this trend even further (Kingma et al., 2014; Salakhutdinov, 2015; Maaløe et al., 2016). As this decade has seen, this revolution comes with its demerits because with more content comes more inaccurate or false content (Balakrishnan et al., 2022; Paschen, 2020; Ozbay and Alatas, 2020). A reliable way to automatically flag these incorrect claims or,

more ambitiously, to automatically correct incorrect claims would do wonders for our ability to manage this risk. Researchers have identified and been working on Factual Claim Verification with some success. This is not the case, however, for Factual Error Correction, where the prohibitively high cost of manually annotating corrections of incorrect claims means there is currently no available dataset for this task (Chen et al., 2022; Thorne and Vlachos, 2021). The few methods that tackle this problem use claim verification datasets for distant supervision and try to use claim verification models to provide signals that can guide the correction process. This exposes the correction methods to flaws in the verification models — one of which is that current verification methods often make either an implicit or explicit domain assumption — focusing on news and political domains (Zeng et al., 2021; Guo et al., 2022). Due to this, today's most potent claim verification methods fail to transfer well to scientific claims, which is especially worrying when we consider that scientific reports and claims are challenging for people without domain expertise to verify or correct. This also adversely impacts the best claim correction methods, with these methods struggling to perform satisfactorily on claim correction tasks from the scientific domain.

In this paper, we introduce $\mathbb{S}$ci$\mathbb{F}$ix, a Factual Claim Correction system that makes no domain assumptions, does not require a claim verification model, and is shown to work well on scientific claims. Our method leverages the power of Large Language Models (LLMs) like GPT (Brown et al., 2020) to generate a claim correction dataset from a claim verification dataset by corrupting 'correct' claims into 'incorrect' ones. This dataset not only allows us to learn a mapping between incorrect claims and their associated correction but also allows us to generate rich annotations via explanations for why this mapping is correct. We use this

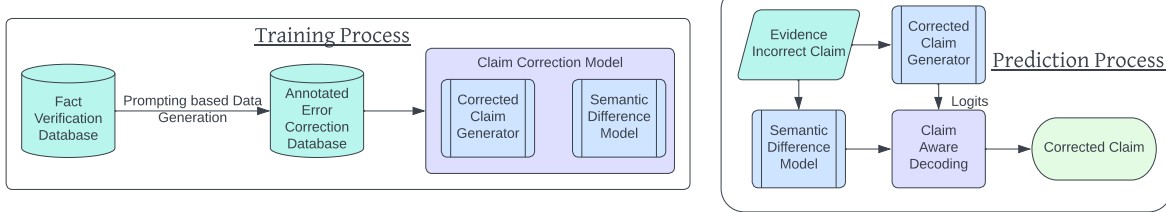

Figure 1: Full Description of the $\mathbb{S}$ci$\mathbb{F}$ix system. During training a fact verification database is converted to a well annotated error correction database using Prompting with LLMs, this database is used to train a Seq2Seq correction LM and a Seq2Label Semantic Difference Model. During prediction the semantic difference model helps guide the generative model using claim aware decoding to generate the corrected claim

dataset to train a conditional generation language model to map evidence and incorrect claims to the appropriate correction, and the explanations serve as a useful guide during the learning process. Finally, we introduce a claim-aware decoding procedure to guide the generation process. While this component does not require a verifier, any verifier can be easily integrated into our procedure if available for the domain at hand. $\mathbb{S}$ci$\mathbb{F}$ix can achieve 84.73% correction accuracy on the SciFact dataset (Wadden et al., 2020a), 77.77% (Wadden et al., 2022) on the SciFact-Open dataset, and 72.75% on the CovidFact dataset, which is an order of magnitude better than competing methods, with the best contemporary method achieving 7.6%, 5% and 15.45% on the datasets, respectively. More impressively, our method outperforms the pretrained LLMs that generated the dataset. Despite using around 800 times as many parameters as our model, Few-Shot Prompting on GPT3.5 achieves 58.74%, 61.11%, and 64.58% on the datasets, respectively, which is consistently outperformed by our method.

Our work presents an alternative route forward for claim correction efforts that do not rely on having access to a powerful verification model and, more generally, shows that general LLMs can be used effectively as part of the model training pipeline to create a more compact yet more powerful model.

In summary, our contributions include:

1. Creation of the $\mathbb{S}$ci$\mathbb{F}$ix system to perform scientific factual error correction,

2. Experiments showing that $\mathbb{S}$ci$\mathbb{F}$ix outperforms current benchmarks, as well as prompting on GPT3, across multiple datasets on both synthetic and naturally generated 'false' claims,

3. Correlation study showing which automatic

metrics are most closely aligned with human evaluation for error correction

## 2 Related Work

**Factual Error Correction:** One of the first approaches for this problem was developed by Shah et al. (2020). Their strategy was to first use a pretrained fact verification model to identify and mask out the components of a sentence that cause the given claim to be incorrect and then use a separate model to fill in the masked sentence while remaining consistent with the provided evidence. Building on these ideas, Thorne and Vlachos (2021) adopts the same masking and infilling paradigm but advances the choice of how to use the fact verification model in the masker and the infilling model. Most recently, Chen et al. (2022) considered fact correction as an iterative sampling problem and sampling editing actions with respect to a target density function. They also used a pre-trained verification model to guide the sampling process. These methods have all made significant advances in factual error correction. However, none are expected to perform reasonably well if their verification model is poor for the task at hand. These methods work around the FEVER dataset, a well-studied fact verification dataset with a suite of powerful verification models that can be plugged into the method. This is not the case in the less studied sub-field of scientific claim verification domains (Wadden et al., 2020b, 2022), where datasets like SciFact-Open prove to be harder to tackle from a verification perspective(Wadden et al., 2022). ZeroFEC (Huang et al., 2023) explores a verifier-free, zero-shot approach to error correction. This method formulates questions about input claims, looks for correct answers in the given evidence, and assesses the faithfulness of each correction based on its consistency

```
Evidence: ...
Incorrect Claim:  A breast cancer patient's capacity to metabolize tamoxifen does not influence
                  treatment outcome.
Correct Claim:    A breast cancer patient's capacity to metabolize tamoxifen  influences
                  treatment outcome.
Explanation:      The evidence mentions that compared with the extensive metabolizers, those with
                  decreased CYP2D6 activity (hetorozygous extensive/intermediate and poor
                  metabolism) had worse event-free survival and desease-free survival,
                  which suggests that capacity to metabolize tamoxifen influences treatment
                  outcome and the claim is true.
```

Figure 2: A sample from the generated dataset

with the evidence.

**Factual Consistency of Summarizations:** Similar problems have arisen in making the summary of a paragraph consistent with the facts of the paragraph. The relevant approaches here are the post-editing processes, which can detect and correct factual errors in the summary. Some methods in this domain are Cao et al. (2020), Kryściński et al. (2019) and Zhu et al. (2020), which try to manually introduce corruptions into correct claims and reconstruct the correct claim using a Seq2Seq model (Sutskever et al., 2014). Our method extends this approach by discarding the need for manually defined entity swapping, back translation, or other labor-intensive methods of introducing corruptions by using an LLM to provide a diverse set of corrupted claims. We also provide a way to generate a set of rich annotations that is fundamentally impossible via the corruption approaches (Cao et al., 2020; Zhang et al., 2022; Zhu et al., 2020).

**Prompting with LLMs:** It has been shown recently that an LLM can achieve high performance on specific few-Shot tasks with a few examples in the context window of the input (Brown et al., 2020; Wei et al., 2022). Chain-of-Thought prompting improves on simple prompting approaches by providing examples in the prompt, which contain question-answer pairs and some stated reasoning for the provided answer. We use these powerful methods with general LLMs to generate data for our more compact and task-dependent model.

## 3 Method

First, we specify how we interpret the claim verification and error correction tasks. Below, we use notation similar to Thorne and Vlachos (2021).

Given a claim $c$ and evidence $E(c)$ such that the evidence contradicts $c$, that is, $E(c) \not\models c$, our task is to generate a corrected claim $c'$, such that $c'$ is supported by the claim: $E(c') = E(c) \models c'$,

and $c'$ makes claims about similar ideas, concepts, objects, etc. as $c$.

In our work, we assume that we have access to a domain-specific claim verification dataset instead of a correction dataset itself. Our system has three key components: i) LLM Driven Data Generation, ii) Domain Adaptive Language Modeling, and iii) a Claim Aware Decoding Procedure.

**LLM Driven Data Generation:** First, we identify something fundamental about the nature of the claim correction problem. One direction, mapping incorrect claims to correct claims, is challenging since it requires a deep understanding of the semantics of both the evidence and claim to perform. However, the reverse direction, mapping correct claims to incorrect claims, is much easier and requires only a partial understanding of the concepts and words in the correct claim, often not requiring any evidence at all. A concrete example of this is shown in Figure 2, where it is possible to generate an incorrect claim from a correct claim without seeing the evidence or understanding any of the medical concepts in the sentence. However, to recover the correct claim, one must comprehend the evidence, identify the error, and correct it. There is a similar pattern in generating explanations for why a claim is valid or rewording a correct claim to maintain the same meaning overall. We exploit this property using a Pretrained LLM (GPT3.5): First, we take the evidence and supported claims from the verification datasets and then produce a correction dataset with annotations and explanations for why the correct claim is true. After these steps, we create an 'augmented correct claim' (an alternate correct claim with the same meaning) for each example. A sample from this dataset is shown in Figure 2, and more are provided in the Appendix Figure 7.

**Domain Adaptive Language Modeling:** We use Domain Adaptive Pretraining (Gururangan et al.,

| Method | SciFact | SciFact-Open | CovidFact | Agreement |
|---|---|---|---|---|
| ZeroFEC | 6.25 ± 3.44 | 5.0 ± 4.33 | 15.45 ± 4.21 | 0.63 |
| VENCE | 7.60 ± 2.41 | 3.72 ± 3.87 | 1.23 ± 7.82 | 0.52 |
| GPT ZS | 54.11 ± 4.92 | 57.73 ± 5.22 | 62.33 ± 4.80 | 0.56 |
| GPT FS | 58.74 ± 2.43 | 61.11 ± 7.85 | 64.58 ± 3.93 | 0.72 |
| $\mathbb{S}$ci$\mathbb{F}$ix Bio | **87.5 ± 2.70** | **80.0 ± 4.21** | 59.25 ± 4.74 | 0.80 |
| $\mathbb{S}$ci$\mathbb{F}$ix All | **84.73 ± 2.71** | **77.77 ± 7.85** | **72.75 ± 4.13** | 0.80 |

Table 1: Mean and standard deviation of correction accuracy as determined by 3 human annotators on a sample of 200 points. The $\mathbb{S}$ci$\mathbb{F}$ix (All) model outperforms all other methods on all datasets and the Bio model increases the margin on the training datasets of SciFact and SciFact-Open while dropping considerably in the hold out CovidFact dataset.

2020) on the evidence passages to give the LLM a better understanding of the kinds of words and concepts that are unique to the specific domain we are interested in. Using this adapted LM, we learn a 'claim correction model'. The claim correction model is trained to map the evidence and incorrect claim to a correct claim and explain why that is a correct claim. Unlike previous methods that interpret error correction as a masking and infilling problem, we interpret it as a conditional generation task. This allows us to tackle a significantly more diverse set of incorrect claims because many incorrect claims are harder to correct by swapping tokens, as they may require the sentence structure to change considerably.

**Claim Aware Decoding Procedure:** Lu et al. (2021) introduced an effective way to perform constrained decoding for generative LLMs. During beam search decoding, instead of scoring a partial sequence as just the probability of generating that sequence of tokens, the constrained decoding method performs a look-ahead search to make a greedy estimate of what the complete sequence is likely to be. Then it incorporates the goodness of the entire sequence into the score of the partial sequence. We utilize the fact that the corrected claim should not have the same meaning as the incorrect claim to guide the decoding process. Specifically, we use the same adapted LM from the previous component to learn a 'semantic difference model'. The semantic difference model is a classifier trained to identify when two claims are semantically similar, giving a score of 1 when they have a different meaning and 0 when they are identical. We then use this as the scoring function for the look-ahead estimate, incentivizing the decoding process to avoid solutions with the same meaning as the incorrect claim.

In the following sections, we showcase the power of this method on scientific claim verification datasets.

## 4 Implementation Details

**Datasets:** We use three different scientific domain claim verification datasets — SciFact, SciFact-Open and CovidFact (Wadden et al., 2020a,b; Saakyan et al., 2021). These datasets come with evidence passages and claims and labels for whether the claim is supported or refuted. Samples can be found in the Appendix (Figure 7). We use only the gold standard evidence paragraphs as input to all methods during training and evaluation.

**Components:** For the LLM Driven Data Generation step, we used few-shot prompting on GPT3.5. For the Domain Adaptive Language Modeling, we trained a T5-base model (Raffel et al., 2020) (220M params) on the abstracts of SciFact and SciFact-Open. After these steps, we fine-tuned this model for the Claim Correction Model and Semantic Difference Model.

**Variation:** We use the components mentioned above to train two models: the first of which is trained on all available datasets ($\mathbb{S}$ci$\mathbb{F}$ix All). For the second model ($\mathbb{S}$ci$\mathbb{F}$ix Bio), we separate our datasets into two distinct 'domains'. SciFact and SciFact-Open are biomedical datasets that use abstracts as their evidence passages. CovidFact is a dataset from news reporting on COVID-19, with selected sentences as evidence passages. We split these into the 'biomed' and 'covid news' domains, respectively, and observe the performance when the training data is taken from the biomed domain, and the prediction task is on the covid news domains. This tests the ability of the methods to generalize to shifting domains. All software implementations are publically available at: `https://github.com/DhananjayAshok/SciFix`

```
Explanation | Incorrect Prediction: The evidence mentions that self-harm rates were
more than ten times higher in female prisoners than in male inmates
Claim: The risk of male prisoners harming themselves is 10 times
that of female prisoners.
```

Figure 3: Explanation directly contradicts the predicted claim, signaling a mistake

| Metric | ZeroFEC | VENCE | GPT ZS | GPT FS | SciFix Bio | SciFix All |
|--------|---------|-------|--------|--------|------------|------------|
| DiffModel | 39.78 | 28.33 | 50.66 | 49.67 | 74.69 | 18.72 |
| SARI | -9.49 | 10.66 | 46.19 | **57.44** | 48.58 | 62.66 |
| GPTScore | **73.68** | **68.14** | **62.21** | 43.83 | **92.45** | **81.12** |
| Annotator 1 | 86.13 | 87.62 | 71.31 | 87.09 | 95.42 | 86.29 |
| Annotator 2 | 85.3 | 90.23 | 87 | 87.92 | 88.95 | 83.75 |
| Annotator 3 | 87.22 | 84.44 | 87.76 | 84.12 | 95.05 | 87.34 |

Table 2: Correlation between average correction accuracy as judged by human annotation and automatic metrics. DiffModel is the output of the Semantic Difference Model, and GPTScore is the average correction accuracy as judged by few-shot prompting on GPT3.5. Results show that on all methods other than FewShot Prompting on GPT, the GPTScore metric consistently has the highest correlation with average correction accuracy

```
Sources: ...
References:  40mg/day dosage of folic acid and 2mg/day dosage of vitamin B12 does not affect
        chronic kidney disease (CDK) progression.
Prediction 1:  40mg/day dosage of folic acid and 2mg/day dosage of vitamin B12 has an affect on
        chronic kidney disease (CDK) progression.
Prediction 2:  40mg/day dosage of folic acid and 2mg/day dosage of vitamin B12 has an affect on
        chronic kidney disease (CDK) progression.

        SARI 1: 92.91      SARI 2: 75.75
```

Figure 4: SARI metric is higher for an incorrect claim than a correct claim. It is unable to identify semantic similarity

## 5 Experiment 1: Correction Dataset

In the first experiment, we use human annotation to measure the accuracy of the claim correction datasets generated from the claim verification datasets. We conduct a survey of using 200 examples from each model, with each example being seen by three distinct annotators. Annotators are asked to select whether the predicted claim is both 1) true given the evidence provided and 2) makes a claim related to the original, incorrect claim. We compute the correction accuracy based on these responses. The annotators were recruited through the personal networks of the authors and were not provided any compensation for the task. To mitigate potential bias in annotation, the annotators were given no information regarding how the predicted claims were generated. Annotators were provided random samples of output from all the models without any indication of which model produced which output, additionally the annotators were not even aware that there could be multiple models responsible for different outputs that they observed. Finally, the annotators were given no details about any of the methods they evaluated, only a description of the claim correction task. The annotation task itself does not involve any graphic descriptions, disturbing language pertaining to sensitive topics (including but not limited to hate speech, violence, sexual content etc.) or personal information. Additionally, all the annotators were shown examples of the task text beforehand and were informed that they were free to decline the task at any time, even after starting. More details available in the appendix (Figure 6).

We use these scores to measure the correlation between human evaluation and several automatic metrics. The metrics used are (i) Semantic Difference Model: the scores of the semantic difference model we trained to use for claim aware decoding, (ii) Few Shot Prompting on GPT3.5: a binary indicator of whether GPT3.5 classifies the prediction as correct or not, see the appendix for the exact prompt used (Figure 9) and (iii) SARI (Xu et al., 2016), the most commonly used metric in the claim correction literature, which has its origins in summarization.

**Baselines:** We compare our method against the following approaches - ZeroFEC (Huang et al., 2023), VENCE (Chen et al., 2022), Zero-Shot Prompting on GPT3.5, and Few-Shot Prompting on GPT3.5 (2 examples) (Brown et al., 2020).

**Results:** Table 1 shows that $\mathbb{S}$ci$\mathbb{F}$ix consistently outperforms all other methods on all datasets, outperforming FewShot Prompting on GPT3.5 by around 16.66% (absolute) on average. The inter-annotator agreement for the methods shows that these results are relatively robust across all annotators. The poor performance of VENCE is possibly explained by the fact that the verifier used is not specialized in scientific error verification; it is possible that if, in the future, a verifier that performs well on all of these datasets is released, it will be able to perform significantly better, however as we noted in the introduction verification methods have yet to perform as well on datasets like SciFact-Open (Wadden et al., 2022). The $\mathbb{S}$ci$\mathbb{F}$ix Bio model can perform better on its training domain of SciFact and SciFact-Open while not generalizing to the unseen covid domain. With a 59% accuracy on CovidFact, however, this method is sufficiently close to the performance of the GPT methods that it shows our method does have some ability to generalize across different domains.

Surprisingly, we find that despite never being explicitly guided to not alter an input claim if it is already true, we can observe this behavior in the model. When the input to our model is the set of all the claims labeled 'SUPPORTED' in the verification datasets (i.e., already true claims), $65.82\%$ of the time the model does not alter the input claim (i.e., string equality with the input claim which was correct), this shows our model is not simply negating or flipping the meaning of input sentences but making more guided corrections to align with the evidence provided.

A qualitative inspection of our model's performance shows that it can make a wide variety of edits and corrections, including swapping incorrect entities, correcting erroneous numerical figures, flipping incorrect adjectives or quantifiers, as well as swapping incorrectly placed concepts(Figures 5, 12). However, the system's mistakes are equally diverse, with no clearly discernable trend in which types of examples are more complex for the model to correct appropriately (See the Appendix for examples Figure 11).

Table 2 shows the correlation of the average correction accuracy with automatic metrics. On $\mathbb{S}$ci$\mathbb{F}$ix ZeroFEC, and VENCE, the GPT-based metric is by far the most correlated with the average correction accuracy, nearly reaching as much correlation as the individual annotator scores themselves on cer-

tain datasets. We find that despite its widespread adoption in the task of error correction, the SARI metric is not as well correlated with correction accuracy. This is perhaps explained by the fact that SARI lacks the ability to detect semantic equivalence and difference as a summarization metric that uses token edit distance between a set of reference sentences and a prediction. We show a motivating example in Figure 4, where a reasonably common example set of sentences with an extremely high SARI score concerning the label can still be incorrect answers. Similarly, a candidate with a lower SARI score can be a more appropriate answer. We report a table with the automatic metrics for each method (Table 3), and the best SARI scores are achieved by our method; however, both ZeroFEC and VENCE have higher SARI scores than both GPT3.5 methods, further suggesting that this metric can be a misleading one when used to benchmark the performance of claim correction systems. The GPT metric is not without its faults; however, as on the GPT methods, it is even less correlated with the correction than SARI and consistently overestimates the performance of the GPT-based methods while underestimating the performance of all other methods. Additionally, due to the black box nature of this metric, we have little way of understanding why this occurs and whether there is a way to mitigate it.

## 6 Experiment 2: Verification Dataset

In the second experiment, we measure the ability of the methods to successfully correct the incorrect claims present in the claim verification dataset itself, and this shows that our results are not only applicable to the specific correction dataset we generated but are more generally applicable to incorrect scientific claims. We use as benchmarks only the best methods from the previous experiments.

**Results:** The results are shown in Table 4 must be interpreted cautiously; the inter-annotator agreement is very low at 0.24, resulting in substantial standard errors on the correction accuracy estimates. This is caused, in part, by the dataset not being the most appropriate for the task of error correction. There are several noisy examples, i.e., supposedly incorrect claims that are actually correct, and more importantly, many examples where it is unclear whether it is possible to 'correct' the claim. However, it is more appropriate to restate the evidence completely (Figure 13). With stan-

dard deviations of this scale, we do not try to claim that any method outperforms the other based on the average correction accuracy. However, we can say that, at the very least, our method is competitive with both Zero-Shot and FewShot Prompting on GPT3.5 on this test set. This experiment helps verify that our method can handle incorrect claims generated through a completely different process than the one used to create our error correction dataset.

## 7 Ablation and Analysis

We next vary components of the pipeline to investigate their individual contribution to the system's success. Specifically, we remove the explanations and the claim-aware decoding procedure from the pipeline. Table 5 shows the results when we use the automatic metric most correlated with human evaluation as judged by Table 2: correction accuracy as computed by FewShot Prompting on GPT3.5. The setting 'all components' is most consistently the best option; however, there are datasets where not having an explanation gives marginally better performance and one dataset where replacing the T5 model with a BART model achieved better performance. The change in the decoding strategy, however, seems to have an apparent adverse effect on the system. The setting with beam search decoding is never better than the claim-aware decoding method. An inspection of the decoding output shows examples where the claim-aware decoding method fails because the semantic difference model is not robust to small meaningless perturbations. As seen in Figure 8, the prediction sentence is semantically the same as the incorrect claim. However, the semantic difference model outputs a high score, implying that the two sentences have different meanings.

The explanations show an interesting trend, with the explanation's coherence and faithfulness being empirically correlated with the correctness of the predicted claim. This can be seen with an example in Figure 3, where an explanation is incoherent, preempting an incorrect prediction. More examples of explanations having this effect are provided in the Appendix Figure 10

## 8 Limitations and Future Work

$\mathbb{S}$ci$\mathbb{F}$ix relies on data generation from LLMs whose training sets are undisclosed to the public (Brown et al., 2020). This can lead to data contamination

| | SciFact | | | SciFact-Open | | | CovidFact | | |
|---|---|---|---|---|---|---|---|---|---|
| | SARI | GPT | Diff | SARI | GPT | Diff | SARI | GPT | Diff |
| VENCE | 77.48 | 2.27 | 13.01 | 82.61 | 0 | 16.69 | 65.62 | 0 | 13.66 |
| ZeroFEC | 83.82 | 1.13 | 8.15 | 84.66 | 16.66 | 18.49 | 78.32 | 15.09 | 15.12 |
| GPT ZS | 66.62 | 72.72 | 50.96 | 81.79 | 83.33 | 66.66 | 71.93 | 70.75 | 64.62 |
| GPT FS | 66.66 | **86.2** | 47.77 | 72.71 | **83.33** | 33.43 | 62.75 | 84.46 | 52.58 |
| Our Best | **94.63** | 77.21 | **92.36** | **94.98** | 80 | **89.99** | **89.77** | 70.27 | **79.97** |

Table 3: Automatic Metrics on the methods. GPT is the FewShot Prompting GPT3.5 correction accuracy estimator, Diff is the average output of the Semantic Difference model. SARI score gives misleadingly high scores to VENCE and ZeroFEC, while the GPT score overestimates the performance of the GPT-based methods

| Method | SciFact | SciFact-Open | CovidFact |
|---|---|---|---|
| GPT ZS | $68.64 \pm 15.39$ | $\mathbf{87.60 \pm 13.5}$ | $42.49 \pm 19.4$ |
| GPT FS | $74.74 \pm 10.7$ | $58.57 \pm 14.8$ | $63.3 \pm 11.66$ |
| $\mathbb{S}ci\mathbb{F}ix$ Bio | - | - | $37.95 \pm 10.32$ |
| $\mathbb{S}ci\mathbb{F}ix$ All | $\mathbf{80.54 \pm 13.58}$ | $84.84 \pm 8.45$ | $\mathbf{65.15 \pm 18.22}$ |

Table 4: Correction Accuracy as determined by Human Annotation on the REFUTED claims from the verification datasets, high standard deviation and low inter annotation agreement (0.24 on average) indicates we can not declare any one method to be superior on this task

```
Claim: general exercise therapy is more effective than Scapular stabilizer exercises in reducing
pain and improving function of the shoulder.
Proposed Correction: Scapular stabilizer exercises are more effective than general exercise therapy
in reducing pain and improving function of the shoulder.

Claim: Severe immunosuppression and also a cytokine storm characterize covid-19 infections
Proposed Correction: Severe immunosuppression and
not a cytokine storm characterize covid-19 infections

Claim: APOE4 expression in iPSC-derived neurons increases AlphaBeta production and
tau phosphorylation, delaying GABA neuron degeneration.
Proposed Correction: APOE4 expression in iPSC-derived neurons increases AlphaBeta production and
tau phosphorylation, promoting GABA neuron degeneration.

Claim: The most prevalent adverse events to Semaglutide are cardiovascular.
Proposed Correction: The most prevalent adverse events to Semaglutide are metabolic.

Claim: Coronavirus vaccine to be tested on mice by june 1
Proposed Correction: Coronavirus vaccine to be tested on humans by june 1

Claim: Uk rolls out all doses of covid-19 vaccination.
Proposed Correction: Uk rolls out first doses of covid-19 vaccination
```

Figure 5: Examples of the claim correction system performing well on a diverse set of input types. The system is able to swap entities, negate only specific parts of a sentence, alter numerical quantifiers in word form, and more

| Ablations | SciFact | SciFact-Open | CovidFact |
|---|---|---|---|
| All Components | 77.21 | **80** | **70.27** |
| No explanations | 73.41 | 80 | 70.27 |
| Beam Decoding | 70.88 | 80 | 65.45 |
| BART Model | **84.81** | 66.66 | **65.6** |

Table 5: The scores stand for the percentage of correction accuracy as computed by Few Shot Prompting on GPT3.5. Ablation over system components shows that the system with all components is the best choice. The effect of the variation of components is highly dependent on the data set.

questions, where the Language Models have in some way seen the testing set. This is not a worry for the SciFact Open dataset (Wadden et al., 2022) as it was released after the update cutoff date for GPT3.5 (Sept 2021). However, it is worth noting that since our comparisons are against GPT3.5, any data contamination concerns affect those baselines far more than it affects our method.

Our method is verifier free. However, there is considerable scope for improvement if we can devise a method to integrate a verifier into the method in a 'soft' way, perhaps with a parameter to balance the tradeoff between how much the method will trust the predictions of the verifier. Additionally, our method does not use the SUPPORTED claims from the claim verification datasets during training. Future work may study how to use this data split to improve our method efficiently. This could make the model robust to incorrect and correct claims inputs. More immediately, our method could be improved by employing prompt engineering to optimize the diversity and quality of the datasets generated and using more powerful Semantic Difference models or score signals during the claim-aware decoding procedure.

There are several interesting directions for future work using this method of leveraging the power of prompting to generate synthetic datasets. There are potentially several problems where the key bottleneck has been the prohibitive cost of human annotation and data creation — the success of this method on Factual Error Correction suggests there may be other subtasks where this method would perform well.

## 9 Conclusion

In this work, we put forward 𝕊ci𝔽ix, a new way to utilize existing Claim Verification datasets to perform Claim Correction effectively. We show that our method is not only more powerful than existing methods on Scientific Claim Correction

but also outperforms Prompting on GPT3 — one of the largest Language Models accessible today.

Factual Claim Correction and Scientific Claim Correction, in specific, are vital yet nascent subfields of NLP. We hope this work expands the range of possible solutions by offering an effective way to perform correction without access to a powerful verification model.

## 10 Acknowledgment

This material is based upon work supported by the U.S. Army Research Office and the U.S. Army Futures Command under Contract No. W911NF-20-D-0002. The content of the information does not necessarily reflect the position or the policy of the government and no official endorsement should be inferred

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

## A    Appendix

### A.1    Details of the Human Annotation Task

For the survey we had both the researchers and multiple personal associates of the researchers participate in the grading. In total we used 200 examples, ensuring that each example is seen by 3 distinct annotators. The annotators were presented the evidence, incorrect claim, correct claim (ground truth), and proposed claim. The annotator is also provided with convenience indicators to show if the proposed claim is exactly equal to the correct or incorrect claim (string inequality) to make annotation easier. The annotator is asked to mask this as either correct (proposed claim changes the incorrect claim into a claim that is true wrt the evidence and also makes a claim about the same topics as the incorrect claim, incorrect: claim incorrect (proposed claim is either the same in meaning as the incorrect claim or is incorrect wrt the evidence in some other way), incorrect: claim correct but unrelated to the original claim (if the proposed claim while technically true does not achieve the intended goal of the claim correction task e.g. a constant output of "COVID-19 is a virus" should not be marked as correct unless it is relevant to the incorrect claim made, this option is also to be selected for trivial answers e.g. completely restating the entire or large parts of the evidence passage). To mitigate potential bias in annotation, the annotators were given no information regarding how the predicted claims were generated. Annotators were provided random samples of output from all the models without any indication of which model produced which output, additionally the annotators were not even aware that there could be multiple models responsible for different outputs that they observed. Finally, the annotators were given no details about any of the methods they evaluated, only a description of the claim correction task. An example HIT is shown in Figure 6

### A.2    Recruitment

The annotators for the task were 3 authors of the paper and 3 personal associates of the authors, hence 6 people overall. While the platform used to perform the annotation was Amazon Mechanical Turk the "hiring" itself did not take place through any platform, no external workers were employed and the personal associates were reached using personal networks of one of the authors. Specifically, the author asked around their friend groups if anyone could volunteer their time and three people said they were available. Neither the authors nor the associates were provided direct monetary compensation for this task.

### A.3    Ethical Considerations

The annotation task itself does not involve any graphic descriptions, disturbing language pertaining to sensitive topics (including but not limited to hate speech, violence, sexual content etc.) or personal information. Additionally all the personal associates were shown examples of the task text beforehand and were informed that they were free to decline the task at any time, even after starting.

Incorrect Claim with Evidence:
Evidence: However, specific ontogenies of individual populations and the overall functional organization of this cellular network are not well defined. Here we report a fate-mapping study of the murine monocyte and macrophage compartment taking advantage of constitutive and conditional CX(3)CR1 promoter-driven Cre recombinase expression. We have demonstrated that major tissue-resident macrophage populations, including liver Kupffer cells and lung alveolar, splenic, and peritoneal macrophages, are established prior to birth and maintain themselves subsequently during adulthood independent of replenishment by blood monocytes. Furthermore, we have established that short-lived Ly6C(+) monocytes constitute obligatory steady-state precursors of blood-resident Ly6C(-) cells and that the abundance of Ly6C(+) blood monocytes dynamically controls the circulation lifespan of their progeny. Claim: Adult tissue-resident macrophages lack a self-renewing capacity.

Correct Claim:
Adult tissue-resident macrophages possess a self-renewing capacity.

Proposed Correction:
Microphages lack self-renewal capacity in adult tissue-resident macrophages.

Proposed Correction is same as the incorrect claim: False

Proposed Correction is same as the correct claim: False

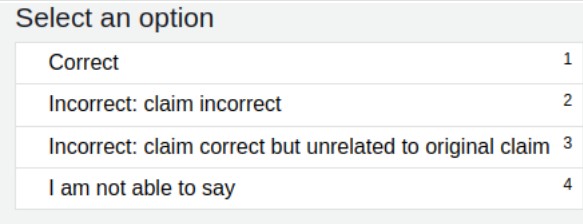

| Select an option | |
|---|---|
| Correct | 1 |
| Incorrect: claim incorrect | 2 |
| Incorrect: claim correct but unrelated to original claim | 3 |
| I am not able to say | 4 |

Figure 6: Example of a HIT for Human annotation

```
Dataset: SciFact
Evidence: Historically, the ribosome has been viewed as a complex ribozyme with
constitutive rather
than regulatory capacity in mRNA translation. Here we identify mutations of the
Ribosomal Protein L38 (Rpl38) gene in mice exhibiting surprising tissue-specific
patterning defects,
including pronounced homeotic transformations of the axial skeleton.
In Rpl38 mutant embryos, global protein synthesis is unchanged; however,
the translation of a select
subset of Homeobox mRNAs is perturbed. Our data reveal that RPL38 facilitates
80S complex formation
on these mRNAs as a regulatory component of the
ribosome to confer transcript-specific
translational control.
Incorrect Claim:  Enhancement of Rpl38 alters the composition of the Hox gene
mRNAs translation in mice without raising overall protein synthesis.
Correct Claim:  Reduction of Rpl38 alters the composition of the Hox gene mRNAs
translation in mice without lowering overall protein synthesis.
Explanation:  The evidence explicitly mentions that in Rpl38 mutant embryos,
global protein synthesis is unchanged,
but the translation of select Hox mRNAs is perturbed,
thereby confirming the fact that reduction of Rpl38 alters
the composition of Hox gene mRNAs translation in mice
without lowering overall protein synthesis and the claim is true.

Dataset: SciFact-Open
Evidence: ...
Incorrect Claim:  Inhibiting glucose-6-phospate dehydrogenase facilitates
lipogenesis by deactivating LKB1-AMPK signaling.
Correct Claim:  Inhibiting glucose-6-phospate dehydrogenase impairs lipogenesis by
reactivating LKB1-AMPK signaling.
Explanation:  The evidence mentions that suppressing 6PGD decreased lipogenesis by
disrupting the active LKB1 complex and reactivating
AMPK activation and that this was mediated via Ru-5-P

Dataset: CovidFact
Evidence: Meanwhile, management of the stock has been passed from a German
owned multinational to an American one. The distribution of that stockpiles PPE,
it seems, may no longer solely be the responsibility of Movianto.
Incorrect Claim:  Private firm running uk ppe stockpile was not sold in middle of
pandemic.
Correct Claim:  Private firm running uk ppe stockpile was sold in middle of pandemic.
Explanation:  The evidence explicitly mentions that the management of the stock was
passed from a
German-owned multinational to an American one, hence the claim is true.
```

Figure 7: Samples from the generated dataset. The first SciFact example shows how sometimes negating multiple words can make a claim that is actually not directly contradicted by the evidence due to double negatives

```
Incorrect Claim: First case of covid-19 reinfection?
Predicted Claim: First report of covid-19 reinduction?
```

Figure 8: Score based decoding failing to prevent a semantically equivalent solution due to fragility of the semantic difference model

```
For the following evidence and claim (fact may or may not be included),
declare whether
the claim is supported by the evidence and fact (TRUE) or not (FALSE):

Evidence: 'Policies requiring discontinuation of methadone in 32% of all
programs contradict the evidence base

for efficacy of long-term replacement therapies and potentially result
in relapse of previously stable patients

Fact: '32% of liver transplantation programs allowed patients to continue
methadone treatment in 2001.'

Claim: '48% of liver transplantation programs allowed patients to continue
methadone treatment in 2001.'
Answer: False

Evidence: Although disorder severity was correlated with probability of
treatment in almost all countries,
35.5% to 50.3% of serious cases in developed countries and 76.3% to 85.4% in
less-developed countries received
no treatment in the 12 months before the interview.
Fact: 76-85% of people with severe
mental disorder receive no treatment in low and middle income countries.
Claim: 76.3-85.4% of people with severe mental disorder receive no treatment in
low and middle income countries.
Answer: True

Evidence: 'Policies requiring discontinuation of methadone in 32% of
all programs contradict the evidence base
for efficacy of long-term replacement therapies and
potentially result in relapse of previously stable patients
Fact: None
Claim: Liver transplantation is a risky procedure
which has a non trivial chance of fatal complications
Answer: False

Evidence: Although disorder severity was correlated with
probability of treatment in almost all countries,
35.5% to 50.3% of serious cases in developed countries and
76.3% to 85.4% in less-developed countries received
no treatment in the 12 months before the interview.
Fact: None
Claim: '35.5% to 50.3% of people with severe mental
disorder receive no treatment in developed countries.
Answer: True
```

Figure 9: Exact prompt given to GPT3.5 for the Correction Accuracy as estimated by FewShot Prompting on GPT3.5

```
Explanation | Incorrect Prediction: The evidence explicitly states that there is no
clear scientific evidence linking ibuprofen and
other NSAIDs to worsening of COVID-19.
Claim: taking medication to lower fever, such as paracetamol (tylenol) and
ibuprofen (advil) worsen COVID-19.

Explanation | Incorrect Prediction: The evidence mentions that human
coronavirus 229E was rapidly inactivated on a range of copper alloys and
that Cu/Zn brasses were very effective at lower copper
concentration, indicating that it can live for hours to days on surfaces
like countertops and doorknobs.
Claim: The coronovirus can live up to 24 hours to months on surfaces
Like countertops or doorkniobs.
How long it survives depends on the material the surface is made from.

Explanation | Incorrect Prediction: The evidence mentions that a detailed analysis
of multiple muscles harvested at various time points during aging showed that the
muscles were of normal size, despite low regenerative capacity, but did
have increased fibrosis, suggesting that lifelong
reduction of satellite cells neither accelerated nor exacerbated sarcopenia.
Claim: Satellite cell dysfunction is a key factor in sarcopenia development

Explanation | Incorrect Prediction: The evidence mentions that RmYN02, the
closest bat virus to SARS-CoV-2, is a recombinant with a structure that includes
differential CpG content in Spike
Claim: Natural selection in the evolution of sars-cov-2 in cattle,
not humans, created a highly capable human pathogen

Explanation | Incorrect Prediction: The evidence mentions that Treg cells lacking
expression of integrin v8 were unable to suppress
pathogenic T cell responses during active inflammation, hence the claim is true.
Claim: T regulatory cells (tTregs) lacking v8 are more adept at
suppressing pathogenic T-cell responses during active inflammation.
```

Figure 10: Explanation directly contradicts the predicted claim, signaling a mistake

```
Incorrect Claim: Who extends restrictions beyond easter
Correct Claim:   Trump extends restrictions beyond easter
Proposed Correction: Who extends restrictions beyond easter

Incorrect Claim: One in two surgical randomized controlled trials
are discontinued early.
Correct Claim:   One in five surgical randomized controlled trials
are discontinued early.
Proposed Correction: One in two surgical randomized controlled trials
are discontinued early.

Incorrect Claim: Deletion of v8 results in a spontaneous inflammatory phenotype.
Correct Claim:   Deletion of v8 has no significant effect on inflammatory phenotype.
Proposed Correction: Deletion of v8 results in a spontaneous inflammatory phenotype.
```

Figure 11: Examples of the claim correction system performing poorly on a diverse set of input types. The system
fails on cases where entities must be switched, quantifiers must be changed and negations must be inserted

```
Claim: general exercise therapy is more effective than Scapular stabilizer
exercises in reducing pain and improving function
of the shoulder.
Proposed Correction: Scapular stabilizer exercises are more effective than general
exercise therapy in reducing pain and
improving function of the shoulder.

Claim: Severe immunosuppression and also a cytokine storm
characterize covid-19 infections
Proposed Correction: Severe immunosuppression and
not a cytokine storm characterize covid-19 infections

Claim: APOE4 expression in iPSC-derived neurons increases AlphaBeta production and
tau phosphorylation, delaying GABA neuron degeneration.
Proposed Correction: APOE4 expression in iPSC-derived neurons increases
AlphaBeta production and tau phosphorylation,
promoting GABA neuron degeneration.

Claim: The most prevalent adverse events to Semaglutide are cardiovascular.
Proposed Correction: The most prevalent adverse events to Semaglutide are metabolic.

Claim: Coronavirus vaccine to be tested on mice by june 1
Proposed Correction: Coronavirus vaccine to be tested on humans by june 1

Claim: Uk rolls out all doses of covid-19 vaccination.
Proposed Correction: Uk rolls out first doses of covid-19 vaccination

Claim: The risk of cancer is lower in individuals with a history of
heavy alcohol consumption
Proposed Correction: The risk of cancer is higher in individuals with a
history of heavy alcohol consumption.
```

Figure 12: Examples of the claim correction system performing well on a diverse set of input types. The system is able to swap entities, negate only specific parts of a sentence, alter numerical quantifiers in word form, and more

```
Evidence: The Expert Working Group on the Commission of Human Medicines in the UK
and other organizations have stated that there is insufficient evidence to
establish a link between ibuprofen and susceptibility
to or exacerbation of COVID-19.
Incorrect Claim: If Fever Helps Fight Infection, Should I Avoid Fever-Reducing Drugs?

Evidence: At the time, no direct evidence existed regarding community spread of
this particular virus, and most previous studies were done in clinical settings.
According to one report, officials were also concerned that widespread masking
would lead to a false sense of security,
leading people to ignore other safety measures,
such as handwashing and self-isolation.
Incorrect Claim: Scientists continue to use common sense early in the pandemic

Evidence: This clinical trial is designed to assess the safety, reactogenicity and
immunogenicity of mRNA-1273 manufactured by ModernaTX, Inc. mRNA-1273 is a
novel lipid nanoparticle (LNP)-encapsulated mRNA-based vaccine that
encodes for a full-length, prefusion stabilized spike (S) protein of SARS-CoV-2.
We aimed to assess the safety and immunogenicity of an inactivated
severe acute respiratory syndrome coronavirus 2 (SARS-CoV-2)
vaccine candidate, BBIBP-CorV, in humans.
Incorrect Claim: Safety and immunogenicity study of c-ncov vaccine to
prevent sars-cov-2 infection

Evidence: Healthcare workers and immunosuppressed or renal patients
had at greater risk of SARS-COV-2 reinfection.
Incorrect Claim: Predictors of symptomatic

Evidence: The company has already developed a lab-based test thats handheld
and gives results in 15 minutes.
Incorrect Claim: Researchers race to develop in-site testing
for covid-19, a potential game changer hospital-confirmed sars-cov-2 reinfection
```

Figure 13: Difficult examples from the verification datasets where it is unclear what the correct answer should be or that the supposedly incorrect claim needs correcting at all. There are many such examples, making comparisons of methods noiser