# OpenReview forum: "The student becomes the master: Outperforming GPT3 on Scientific Factual Error Correction"
_EMNLP/2023/Conference — EMNLP 2023 Findings_

### Official Review · Reviewer_eYY6 · 2023-08-03

**Soundness:** 4

**Excitement:**

4: Strong: This paper deepens the understanding of some phenomenon or lowers the barriers to an existing research direction.

**Paper Topic And Main Contributions:**

The paper proposes a scientific error correction pipeline based on Large Language Models (LLMs). In doing so, it generates a factual error correction dataset from a factual claim verification dataset by converting correct claims into false claims and annotating them with explanations. This claim-explanation pair is fed into GPT3.5 to create the required dataset. Then, a language model with claim-aware decoding on top (fine-tuned T5) is used for full supervision. The experimental results show that the proposed system (SciFix) outperforms the existing work by a large margin. It is even better than GPT3.5 for the factual error correction task.

**Reasons To Accept:**

A novel way of constructing factual error correction datasets from factual error verification counterparts.
A novel pipeline (a language model encoder with a custom decoder) for scientific error correction.
The claim-aware decoding improves the correction task by ensuring a semantic difference between false and correct claims.
The authors conduct extensive experiments (including qualitative analysis and ablation study) to support their claims.

**Reasons To Reject:**

-

**Reproducibility:**

4: Could mostly reproduce the results, but there may be some variation because of sample variance or minor variations in their interpretation of the protocol or method.

**Reviewer Confidence:**

4: Quite sure. I tried to check the important points carefully. It's unlikely, though conceivable, that I missed something that should affect my ratings.

---

> ### Author Rebuttal · Authors · 2023-08-26
>
> We thank the reviewer for their comments.

---

### Official Review · Reviewer_hZmU · 2023-08-05

**Soundness:** 3

**Excitement:**

3: Ambivalent: It has merits (e.g., it reports state-of-the-art results, the idea is nice), but there are key weaknesses (e.g., it describes incremental work), and it can significantly benefit from another round of revision. However, I won't object to accepting it if my co-reviewers champion it.

**Paper Topic And Main Contributions:**

The paper presents a factual error correction system SciFix, which is a specialized factual error correction system for scientific domain. To develop such system, the paper proposes a way to generate false claims by using LLMs such as ChatGPT (GPT-3.5) by corrupting the original claim through the designed prompt. By using the annotated dataset from LLMs, the paper train a factual error correction system for scientific claim. Also, the paper leverages claim verification systems into the decoding stage by adopting the previous approach. The proposed factual error correction system achieves new state-of-the-arts in scientific error correction task.

**Questions For The Authors:**

- What is the special ability of LLMs in generating false claims? As shown in Figure 2, incorrect claim just differs in negation, which can be easily generated by using rule-based approaches. I expect some key properties such as statistics of the generated false claims generated by LLMs when I read the paper. In L373, the paper said that the proposed system can make diverse corrections as shown in Figure 4. But, I'm not sure about the type of the errors the model can fix by watching the examples in Figure 4.

**Reasons To Accept:**

- The paper proposes a new method to generate datasets for scientific factual error correction task by leveraging large language models for generating false claims.

**Reasons To Reject:**

- The proposed method seems domain-agnostic, but the paper only focuses on scientific factual error corrections. It would be better to experiment on FEVER dataset that is more common in this area.
- I have some doubts about the reason why the LLMs are useful for generating factually incorrect claims. I expect some explanations on the reason for such ability compared to the previous approaches such as entity replacement and negation.

**Reproducibility:**

4: Could mostly reproduce the results, but there may be some variation because of sample variance or minor variations in their interpretation of the protocol or method.

**Reviewer Confidence:**

4: Quite sure. I tried to check the important points carefully. It's unlikely, though conceivable, that I missed something that should affect my ratings.

**Typos Grammar Style And Presentation Improvements:**

- It would be better to include some analysis(e.g. statistics) on the types of the errors (may be the generated errors from LLMs) that the system can correct.

---

> ### Author Rebuttal · Authors · 2023-08-26
>
> We appreciate the helpful feedback and suggestions. Specifically we thank the reviewer for pointing out how statistical analysis of the errors would be of interest, we will try to work that into the paper.
>
> To address the concerns highlighted, we would first like to mention that the scope of this paper was intentionally limited to Scientific Error Correction (which is why we did not show results on the FEVER dataset). The primary reason we chose to restrict the field in this way is we feel that scientific error correction is a vital domain with some special considerations (e.g. rare vocabulary, different sense of correctness when compared to the more political news setting of FEVER, lack of a strong verification method for datasets like SciFact-Open etc.) which warranted an exclusive focus. It is true that our method does not make any domain assumptions, however we preferred to keep the usage of this method on other more standard claim verification/correction datasets to a different work.
>
> To answer the reviewers question on why LLMs are uniquely useful for generating factually incorrect claims; there are two key reasons we prefer LLMs in generating false claims.
> The first is that many rule based correction systems are not able to handle arbitrary input claims. Specifically in the baselines we implemented, negation based systems fail when the words to be negated were irregular or the direct negation of a word is not an appropriate substitution.  For example: our NLTK rule based baseline which consisted of both entity substitution and negation was unable to find the negation for the word "characterize" in "Severe immunosuppression and also a cytokine storm characterize covid-19 infections", however our LLM method correctly replaces this with "are not associated with". The negation system also converted "Satellite cell dysfunction is a key factor in sarcopenia development" to "Stationary cell ....." which is not a sensible replacement).
>
> Negation systems are also prone to double or multiple negation, which can unintentionally preserve the meaning of the input claim. Generic entity replacement systems fail when the named entities are specific to the scientific domain, e.g. "Satellite cell dysfunction is a key factor in sarcopenia development" has no named entities of the typical (PER, LOC, ORG, MISC types). If we make additional domain assumptions (which we would like to avoid) and use a biomedical NER based substitution system we may identify "sarcopenia" as an entity, but there is no way to replace it with a word that ensures the incorrect claim still makes logical sense. This is important as we do not want our generated incorrect claims to be trivially false. Our NLTK rule based negation and entity replacement system was able to alter only 28% of the datapoints from our various datasets, while prompting on GPT3.5 is able to alter all of them.
>
> The second, more important reason, is that while rule based negation and entity swapping will never be able to change the structure of the sentences, LLMs are able to completely rewrite a sentence. Recent work seems to suggest that both entity replacement and negation insertion result in simplistic generations that do not exhibit linguistic diversity. These minor alterations frequently fall short in offering adequate inductive bias to foster robustness (Khashabi et al., 2020; Huang et al., 2020; Joshi and He, 2022). For example, in our dataset, we have the claim: "Chronic aerobic exercise alters endothelial function, impairing vasodilating mechanisms mediated by NO.", the LLM method generated an equivalent claim: "Endothelial function changes due to chronic aerobic exercise, which weakens the ability of NO to widen blood vessels." and a negated, incorrect claim: "Endothelial function remains stable despite chronic aerobic exercise, which strengthens the capacity of NO to dilate blood vessels." This example shows how the LLM is able to generate equivalent and incorrect claims with greatly altered word placement and sentence structure when compared to rule based approaches. This will enable the creation of much more diverse claims, and when used for training makes it less likely that the model will overfit due to undue focus on just the few tokens or parts of a sentence that have been altered.

---

### Official Review · Reviewer_LL4j · 2023-08-08

**Soundness:** 4

**Excitement:**

3: Ambivalent: It has merits (e.g., it reports state-of-the-art results, the idea is nice), but there are key weaknesses (e.g., it describes incremental work), and it can significantly benefit from another round of revision. However, I won't object to accepting it if my co-reviewers champion it.

**Missing References:**

"Evaluating the Factual Consistency of Abstractive Text Summarization" - Kryscinski et al. is another relevant work that could be cited in the context of generating synthetic data with simple claim transformations in the context of factual correctness (verification rather than correction).

**Paper Topic And Main Contributions:**

In this work the authors propose a new framework for factual claim correction in the context of scientific claims (limited to the bio and med domains). The work introduces three key components: 1) leveraging existing claim verification datasets and available LLMs (GPT) the authors generate a claim correction dataset by using the LLM to create incorrect claims based on correct ones, 2) domain adaptation of the claim correction model, which is based on a T5 LM, by pre-training the model on evidence passages from the claim verification/correction dataset, 3) a claim aware decoding procedure which alternates the scoring function of beam search decoding to include correctness scores for sequence completions found using look-ahead search. Authors conduct the study on 3 datasets: SciFact, SciFact-Open, and CovidFact.  The work also includes a correlation study of available automatic metrics and human annotations.

**Questions For The Authors:**

The first experiments describe the accuracy of the claim correction datasets, what is the accuracy of a dataset?

**Reasons To Accept:**

+ Work in a very relevant domain of factual claim correction
+ Results show promising improvements over previously introduced methods.

**Reasons To Reject:**

- A large part of this method boils down to synthetic data generation, which has been done many times, and has limited novelty for the community
- Despite being a rather straightforward method (which is not a negative), the writing (presentation) seems convoluted and hard to follow

**Reproducibility:**

4: Could mostly reproduce the results, but there may be some variation because of sample variance or minor variations in their interpretation of the protocol or method.

**Reviewer Confidence:**

3: Pretty sure, but there's a chance I missed something. Although I have a good feel for this area in general, I did not carefully check the paper's details, e.g., the math, experimental design, or novelty.

---

> ### Author Rebuttal · Authors · 2023-08-26
>
> Thank you for reviewing our paper and for your feedback. We especially appreciate the highlighting of other relevant work; we will add that to the paper and try to make the writing easier to follow. To answer the reviewer's question on what the accuracies of experiment 1 are:
>
> The accuracy reported in Table 1 is the correction accuracy of a model on a given dataset. This is the percentage of incorrect claims from a given dataset that a model is able to properly "correct" as judged by human annotation. As mentioned in the evaluation subsection of section 4, a predicted claim must be supported by, or true with respect to, the evidence provided as well as make a claim that is related to the originally incorrect claim, to be considered as properly "corrected". The Appendix section A.1 elaborates that the specification that the new corrected claim must be related to the originally incorrect claim is to penalize models that output constant and trivially true claims hence not really correcting the specific claim it was given.
>
> For example, if the incorrect claim is "A breast cancer patient's capacity to metabolize tamoxifen does not influence treatment outcome" and the associated ground truth label is "A breast cancer patient's capacity to metabolize tamoxifen influences treatment outcome", then if the prediction is "breast cancer affects individuals with breasts" it will not be marked as "correct" by the human annotator, but if the prediction is "the ability of breast cancer patients to metabolize tamoxifen has an affect on treatment outcome" it would be marked as "correct".

---

### Official Review · Reviewer_RUuW · 2023-08-09

**Soundness:** 4

**Excitement:**

3: Ambivalent: It has merits (e.g., it reports state-of-the-art results, the idea is nice), but there are key weaknesses (e.g., it describes incremental work), and it can significantly benefit from another round of revision. However, I won't object to accepting it if my co-reviewers champion it.

**Justification For Ethical Concerns:**

This paper hired human annotators to help verify the generated claim. However, it does not contain an ethics statement section in the paper. Although in Appendix A, the authors listed a detailed description of the annotation task, it does not release any details about the hiring process. e.g., how the annotators were hired, by Amazon Turker or other platforms? Furthermore, is this annotation task approved by the institution's IRB?  Therefore, there is an ethical concern.

**Paper Topic And Main Contributions:**

They introduce a scientific claim correction system that makes no domain assumptions and does not require a verifier but is able to outperform existing methods by an order of magnitude.

1. They create the SciFix system to perform scientific factual error correction.
2. Experiments showing that SciFix outperforms current benchmarks, as well as prompting on GPT3, across multiple datasets on both synthetic and naturally generated ‘false’ claims.
3. The correlation study shows which automatic metrics are most closely aligned with human evaluation for error correction.

**Reasons To Accept:**

1. The paper is well-written and easy to follow.

2. The paper did comprehensive experiments and thorough analysis.

3. The paper proposed a strong model for factual error correction. The performance of the proposed system is significantly higher than previous baselines.

**Reasons To Reject:**

1. Missing annotator hiring details: Several details regarding the recruitment of human annotators are missing. Although Appendix A contains a description of the annotation task, it lacks information about the hiring process. For instance, how many annotators were hired in total? Were they recruited through Amazon Turk or other platforms? What was the wage offered? Is this annotation task approved by the IRB?

2. Adoption of distillation idea: This paper has adopted the concept of distilling information from GPT-3 to construct synthetic datasets for training smaller downstream models. While this concept has been used in many previous papers across different tasks, it raises doubts about the unique technical contribution of this paper.

**Reproducibility:**

4: Could mostly reproduce the results, but there may be some variation because of sample variance or minor variations in their interpretation of the protocol or method.

**Reviewer Confidence:**

3: Pretty sure, but there's a chance I missed something. Although I have a good feel for this area in general, I did not carefully check the paper's details, e.g., the math, experimental design, or novelty.

---

> ### Author Rebuttal · Authors · 2023-08-26
>
> We thank the reviewer for their thoughtful comments. We first wanted to address the ethics concerns. The annotators for the task were 3 authors of the paper and 3 personal associates of the authors, hence 6 people overall.  While the platform used to perform the annotation was Amazon Mechanical Turk the "hiring" itself did not take place through any platform, no external workers were employed and the personal associates were reached using personal networks of one of the authors. Specifically, the author asked around their friend groups if anyone could volunteer their time and three people said they were available. Neither the authors nor the associates were provided direct monetary compensation for this task. After the annotation was complete the author treated the non-author annotators to dinner. The annotation task itself does not involve any graphic descriptions, disturbing language pertaining to sensitive topics (including but not limited to hate speech, violence, sexual content etc.) or personal information.  Additionally all the personal associates were shown examples of the task text beforehand and were informed that they were free to decline the task at any time, even after starting. Due to these factors and the nature of the annotation task, we felt it was appropriate to conduct the annotation without review from an IRB.
>
> We also wanted to address the doubts regarding uniqueness of the technical contribution of the paper due to its usage of a distillation idea. This work is undoubtedly inspired by methods that use LLMs to construct synthetic datasets, however there are some unique aspects of this that we feel are unique technical contributions. These include:
> 1. The identification of sentence negation and rewording as subtasks that can be reliably performed by an LLM even without a complete understanding of the evidence or claim itself. Even if one decides to construct a synthetic dataset, it is unclear whether there is a reliable way to produce high quality synthetic datapoints. This work identifies that while claim correction is hard for LLMs to perform without understanding the evidence or scientific concepts in the claim, corrupting a correct claim can be done with just a partial understanding of the words in the sentence.
> 2. Rich annotation: we do not simply construct a dataset of (incorrect claim, correct claim) pairs, but we also generate explanations that prove to be useful for regularization of the learned models. This can still be considered "distilling information from GPT-3", but it is a very specific way to generate high quality annotated data that is not available through any other means (no equivalent dataset exists for scientific or factual error correction)
> 3. Score based decoding: apart from using the dataset to perform supervised learning, we also use it to train a model that can help guide our decoding process and incorporate claim information during this process.
>
> We adopt the general approach of using GPT-3 to extract a dataset, but do so in a specific way to solve unique problems of the Error Correction task, and to enable components (explanations, score based decoding) that are otherwise impossible to create. As such we do not believe the adoptation of distillation undermines the unique technical contributions of the paper.

---

### Meta-Review · Area_Chair_j5B7 · 2023-09-15

**Recommendation:** 3

**Metareview:**

The paper introduces SciFix, a novel scientific error correction system optimized for the bio and med domains, utilizing Large Language Models (LLMs) GPT3.5. By leveraging factual claim verification datasets, the research creates a unique factual error correction dataset. This is achieved by transforming correct claims into false ones using LLMs and subsequently annotating them with explanations. Furthermore, a domain-adapted T5 language model is employed, which incorporates claim-aware decoding to ensure accurate error correction. Notably, this approach surpasses existing benchmarks, even outperforming GPT3.5 in factual error correction tasks. The study encompasses multiple datasets, including SciFact, SciFact-Open, and CovidFact, and delves into the correlation between automatic metrics and human evaluations.

The paper appears solid and worthy of publication. Several reviewers have mentioned that its technical novelty is marginal. Another concern raised by a reviewer and ethics char is the description of hiring annotators, but this can be addressed easily by authors.

---

### Decision · Program_Chairs · 2023-10-07

**Decision:**

Accept-Findings

**Comment:**

The paper introduces SciFix, a novel scientific error correction system optimized for the bio and med domains, utilizing Large Language Models (LLMs) GPT3.5. By leveraging factual claim verification datasets, the research creates a unique factual error correction dataset. This is achieved by transforming correct claims into false ones using LLMs and subsequently annotating them with explanations. Furthermore, a domain-adapted T5 language model is employed, which incorporates claim-aware decoding to ensure accurate error correction. Notably, this approach surpasses existing benchmarks, even outperforming GPT3.5 in factual error correction tasks. The study encompasses multiple datasets, including SciFact, SciFact-Open, and CovidFact, and delves into the correlation between automatic metrics and human evaluations.

The paper appears solid and worthy of publication. Several reviewers have mentioned that its technical novelty is marginal. Another concern raised by a reviewer and ethics char is the description of hiring annotators, but this can be addressed easily by authors.